# A Milky Way-like barred spiral galaxy at a redshift of 3

Luca Costantin[1 ✉], Pablo G. Pérez-González[1], Yuchen Guo[2], Chiara Buttitta[3,4], Shardha Jogee[2], Micaela B. Bagley[2], Guillermo Barro[5], Jeyhan S. Kartaltepe[6], Anton M. Koekemoer[7], Cristina Cabello[8,9], Enrico Maria Corsini[4,10], Jairo Méndez-Abreu[11,12], Alexander de la Vega[13], Kartheik G. Iyer[14], Laura Bisigello[4,10], Yingjie Cheng[15], Lorenzo Morelli[16], Pablo Arrabal Haro[17], Fernando Buitrago[18,19], M. C. Cooper[20], Avishai Dekel[21], Mark Dickinson[17], Steven L. Finkelstein[2], Mauro Giavalisco[15], Benne W. Holwerda[22], Marc Huertas-Company[11,12,23,24], Ray A. Lucas[7], Casey Papovich[25,26], Nor Pirzkal[27], Lise-Marie Seillé[28], Jesús Vega-Ferrero[18], Stijn Wuyts[29] & L. Y. Aaron Yung[30]

The majority of massive disk galaxies in the local Universe show a stellar barred structure in their central regions, including our Milky Way[1,2]. Bars are supposed to develop in dynamically cold stellar disks at low redshift, as the strong gas turbulence typical of disk galaxies at high redshift suppresses or delays bar formation[3,4]. Moreover, simulations predict bars to be almost absent beyond $z = 1.5$ in the progenitors of Milky Way-like galaxies[5,6]. Here we report observations of ceers-2112, a barred spiral galaxy at redshift $z_{phot} \approx 3$, which was already mature when the Universe was only 2 Gyr old. The stellar mass ($M_\star = 3.9 \times 10^9 M_\odot$) and barred morphology mean that ceers-2112 can be considered a progenitor of the Milky Way[7-9], in terms of both structure and mass-assembly history in the first 2 Gyr of the Universe, and was the closest in mass in the first 4 Gyr. We infer that baryons in galaxies could have already dominated over dark matter at $z \approx 3$, that high-redshift bars could form in approximately 400 Myr and that dynamically cold stellar disks could have been in place by redshift $z = 4-5$ (more than 12 Gyrs ago)[10,11].

The barred nature of ceers-2112 (right ascension = 214.97993 degrees; declination = 52.991946 degrees; J2000.0) is identified through the multiwavelength analysis of the James Webb Space Telescope Near Infrared Camera (JWST/NIRCam) images taken during the first epoch (21–22 June 2022) of the Cosmic Evolution Early Release Science (CEERS[12]) campaign. The galaxy was not classified as barred during a visual inspection of the CEERS sample[13], owing to its low surface brightness in the outer regions, especially at short wavelengths where the stellar disk is barely detected. But, at longer wavelengths, ceers-2112 resembles a spiral disk galaxy and the bar component is clearly detected by analysing the composite image obtained by stacking all seven point-spread-function-convolved (PSF-convolved) NIRCam images (Fig. 1a).

The first piece of evidence for the presence of a stellar bar in ceers-2112 is provided by the strong residuals obtained from modelling

the galaxy with a Sérsic component (Fig. 1b). Our findings highlight prominent features in correspondence of the spiral arms and edges of the bar, which reveals that one morphological component is not enough to account for the complex structure of the galaxy (for example, twist of isophotes at small galactocentric distances and strong residuals). Thus, we performed a multicomponent two-dimensional (2D) photometric decomposition of ceers-2112, assuming that its surface-brightness distribution is the sum of a double-exponential disk and a Ferrers bar (Fig. 1c) and found that the galaxy has a stellar bar with length $r_{Ferrers} = 0.42 \pm 0.03$ arcsec (3.3 kpc). The decomposition of the azimuthal luminosity surface-density distribution into the Fourier $m$-components using the composite ceers-2112 image provided the third piece of evidence that the galaxy has a prominent bar (maximum relative amplitude of the $m = 2$ to $m = 0$ component $I_2/I_0 > 0.4$)

[1]Centro de Astrobiología (CAB), INTA-CSIC, Torrejón de Ardoz, Madrid, Spain. [2]Department of Astronomy, The University of Texas at Austin, Austin, TX, USA. [3]INAF - Osservatorio Astronomico di Capodimonte, Napoli, Italy. [4]Dipartimento di Fisica e Astronomia "G. Galilei", Università di Padova, Padova, Italy. [5]Department of Physics, University of the Pacific, Stockton, CA, USA. [6]Laboratory for Multiwavelength Astrophysics, School of Physics and Astronomy, Rochester Institute of Technology, Rochester, NY, USA. [7]Space Telescope Science Institute, Baltimore, MD, USA. [8]Departamento de Física de la Tierra y Astrofísica, Fac. CC. Físicas, Universidad Complutense de Madrid, Madrid, Spain. [9]Instituto de Física de Partículas y del Cosmos (IPARCOS), Fac. CC. Físicas, Universidad Complutense de Madrid, Madrid, Spain. [10]INAF - Osservatorio Astronomico di Padova, Padova, Italy. [11]Departamento de Astrofísica, Universidad de La Laguna, La Laguna, Spain. [12]Instituto de Astrofísica de Canarias, La Laguna, Spain. [13]Department of Physics and Astronomy, University of California, Riverside, CA, USA. [14]Columbia Astrophysics Laboratory, Columbia University, New York, NY, USA. [15]University of Massachusetts Amherst, Amherst, MA, USA. [16]Instituto de Astronomía y Ciencias Planetarias, Universidad de Atacama, Copiapó, Chile. [17]NSF's National Optical-Infrared Astronomy Research Laboratory, Tucson, AZ, USA. [18]Departamento de Física Teórica, Atómica y Óptica, Universidad de Valladolid, Valladolid, Spain. [19]Instituto de Astrofísica e Ciências do Espaço, Universidade de Lisboa, Lisbon, Portugal. [20]Department of Physics and Astronomy, University of California, Irvine, CA, USA. [21]Racah Institute of Physics, The Hebrew University of Jerusalem, Jerusalem, Israel. [22]Physics and Astronomy Department, University of Louisville, Louisville, KY, USA. [23]Université Paris-Cité, Paris, France. [24]Center for Computational Astrophysics, Flatiron Institute, New York, NY, USA. [25]Department of Physics and Astronomy, Texas A&M University, College Station, TX, USA. [26]George P. and Cynthia Woods Mitchell Institute for Fundamental Physics and Astronomy, Texas A&M University, College Station, TX, USA. [27]ESA/AURA Space Telescope Science Institute, Baltimore, MD, USA. [28]Aix Marseille Univ, CNRS, CNES, LAM, Marseille, France. [29]Department of Physics, University of Bath, Claverton Down, Bath, UK. [30]Astrophysics Science Division, NASA Goddard Space Flight Center, Greenbelt, MD, USA. ✉e-mail: lcostantin@cab.inta-csic.es

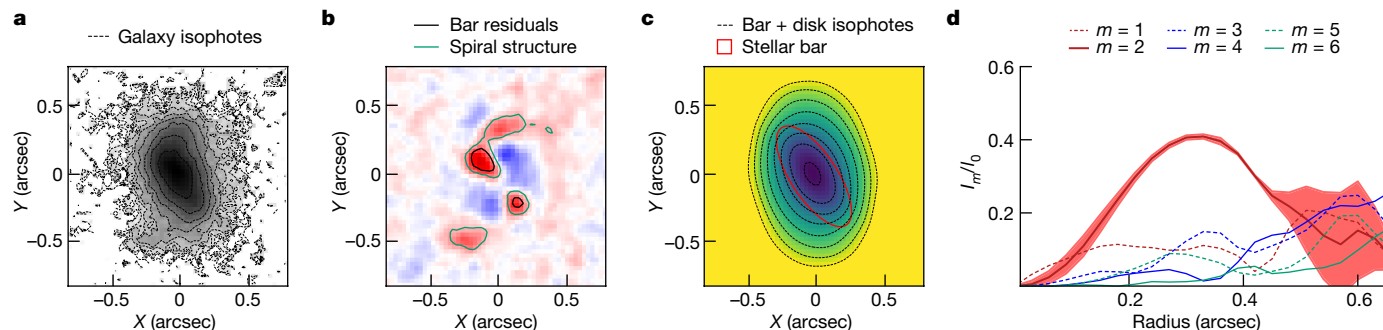

**Fig. 1 | Morphological modelling of ceers-2112. a,** Combined stack image of ceers-2112, with isophotal contours showing an elongated barred structure in the inner region and spiral arms departing from it. **b,** One-component Sérsic residuals, which highlight the bar and spiral structures (black and green contours, respectively). **c,** Two-dimensional bar + disk model, which shows a stellar bar of length $r_{Ferrers} = 0.42 \pm 0.03$ arcsec (3.3 kpc). The bar component is shown as a red solid line and the bar + disk isophotes are shown as black dashed contours. **d,** Radial profiles of the relative amplitude of the odd (dashed lines) and even (solid lines) Fourier components, derived from the deprojected stack image of ceers-2112. The $m = 2$ mode shows a prominent bar (maximum $I_2/I_0 > 0.4$) with strength $S_{bar} = 0.23 \pm 0.01$. Shaded region represents $1\sigma$ confidence interval for the $m = 2$ mode.

with strength $S_{bar} = 0.23 \pm 0.01$ (refs. 14,15). The $m = 2$ peak (Fig. 1d) uniquely describes the barred elongated structure and allowed us to rule out the possibility that the bar could be misinterpreted as spiral arms departing from a compact bulge[16].

Combining the Hubble Space Telescope Advanced Camera for Surveys (HST/ACS), Hubble Space Telescope Wide Field Camera 3 (HST/WFC3) and JWST/NIRCam datasets, we carefully measured ceers-2112 photometry and derived that the galaxy has a photometric redshift of $z_{phot} = 3.03^{+0.04}_{-0.05}$. Taking advantage of the unprecedented spatial resolution, wide wavelength coverage and depth provided by JWST observations, combined with HST datasets, we also derived the 2D spectral energy distribution (SED) of ceers-2112 (ref. 17). We inferred the galaxy star formation history (SFH) from detailed SED fitting (Fig. 2a) and found that it has a total stellar mass of $M_\star = 3.9 \times 10^9 M_\odot$ and a mass-weighted age of $620^{+150}_{-160}$ Myr (Fig. 2b). By comparing ceers-2112 with the assembly history of Milky Way progenitors (Fig. 3), we demonstrated that it can be considered the furthest progenitor of the Milky Way both in terms of structure and assembly history[9,18]. This analysis suggests that the stellar disk of ceers-2112 assembled at $z \approx 5$ and that the bar component formed 200 Myr later, assembling in about 400 Myr, which provides an observational hint on the formation timescale of bars and spiral structures at these early times. The stellar density map built from the spatially resolved stellar population analysis (Fig. 2c) provides an additional independent confirmation of the presence of a stellar bar component, which has $\log(\Sigma) \approx 8.4 M_\odot$ kpc$^{-2}$.

The observational discovery of barred galaxies at $z > 2$ (ref. 19), such as ceers-2112, has strong implications for our understanding of galaxy evolution, in particular, in the first gigayears after the Big Bang. On the one hand, it implies that dynamically cold stellar disks could have formed when the Universe was only a few gigayears old; on the other hand, it puts strong constraints on the dark matter distribution in these galaxies (with baryons dominating over dark matter).

Lambda cold dark matter (ΛCDM) models predict that galaxies at $z > 5$ experienced a phase of gas accretion, forming stars at a very high pace and sustaining the growth of black holes[20,21]. The baryonic cycle of this turbulent phase is balanced by strong outflows due to feedback from active galactic nuclei and supernovae[22,23]. In the state-of-the-art cosmological simulations, different feedback implementations are able to efficiently disperse baryons over large radial scales. However, to build up cold stellar disks and barred galaxies at $z \gtrsim 3$, and Milky Way systems such as ceers-2112 (Fig. 3), models should be able to reproduce baryon-dominated disks with $M_\star < 10^{10} M_\odot$ and net rotation at early times. Recently, it has been shown that some massive disk galaxies ($M_\star > 10^{10} M_\odot$) in the TNG50 cosmological simulation could have been present as early as $z \approx 4$ and that bars could have already started forming at those times[16]. However, despite these findings, cosmological simulations still struggle to produce barred galaxies beyond $z > 1.5$, especially at lower masses[5,6,24].

Owing to their low entropy, galaxy disks with highly ordered rotation are very sensitive to perturbations. However, high-$z$ galaxies are more gas-rich (and turbulent) than local galaxies[3,25–27] and gas-rich stellar disks stay near-axisymmetric much longer than gas-poor ones, which prevents or delays the formation of the bar component[28]. Because ceers-2112 has a mass-weighted age of approximately 600 Myr, the

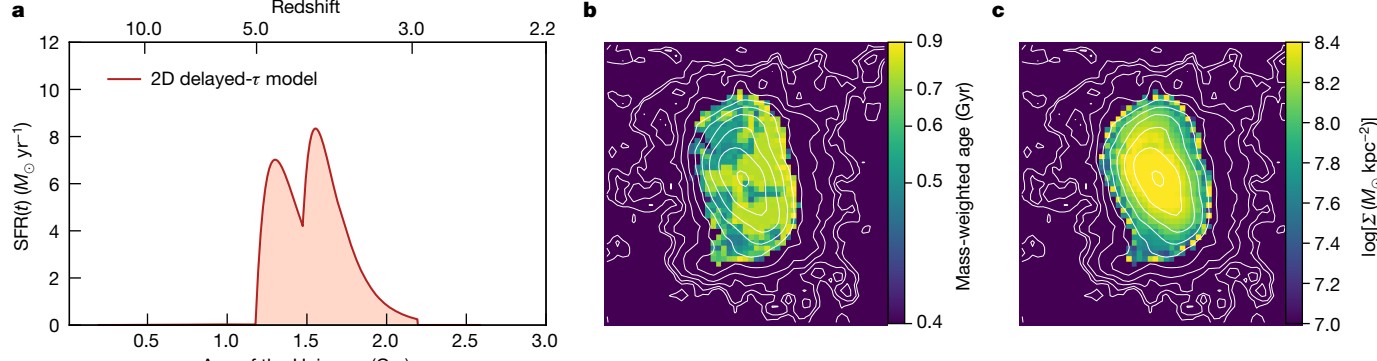

**Fig. 2 | Stellar population properties of ceers-2112. a,** Fiducial spatially resolved SFH derived with synthesizer (delayed-$\tau$ model). **b,** Two-dimensional mass-weighted age map of ceers-2112. **c,** Stellar mass-density map of ceers-2112. The isophotal contours of the stack image are superposed on the mass-weighted age and mass-density maps. Maps in **b** and **c** are 53 × 53 px$^2$, which corresponds to 1.59 × 1.59 arcsec$^2$ (12.5 × 12.5 kpc$^2$ at $z = 3.03$).

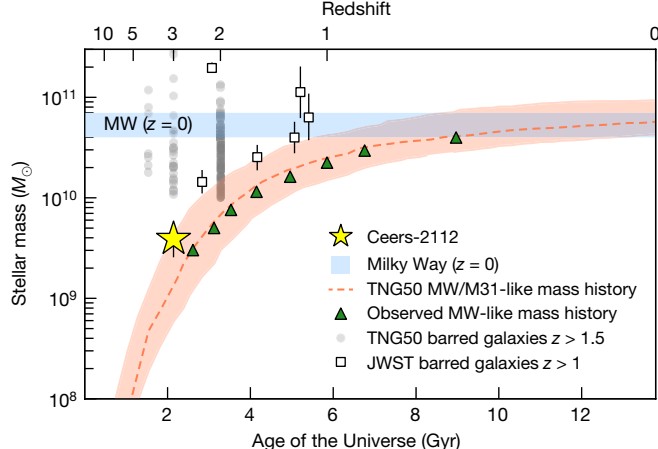

**Fig. 3 | Mass-assembly history of ceers-2112.** Mass-assembly history of Milky Way (MW) and M31 analogues (dashed red line), compared with ceers-2112 (yellow star). The observed stellar masses of Milky Way progenitors are shown as green triangles[7]. The red shaded region shows the 25th–75th percentile range[8]. The blue shaded region stands for the observational estimates for the stellar mass at $z = 0$ of the Milky Way[30–32]. Barred galaxies at $z > 1$ (refs. 19,33) are shown as empty squared symbols and TNG50 predictions of barred galaxies at $z > 1.5$ (ref. 16) are shown as grey dots. Error bars show the systematic uncertainties related to the assumptions of the SFH modelling.

high gas fraction usually observed in high-$z$ galaxies could have been rapidly consumed during the stellar disk growth before the stellar bar component could start developing[29]. Thus, this result highlights the need to investigate the interplay between gas abundance and star formation efficiency in disk galaxies at $z > 2$, which will be fundamental in constraining the formation timescale of bars and the early evolution of disk galaxies. Our findings allow us to speculate that ceers-2112 went through a fast episode of gas consumption when the Universe was only approximately 2 Gyr old ($z ≈ 4$; Fig. 2) that allowed the stellar disk to become dynamically cold and unstable enough to allow a bar to form and grow in less than 400 Myr (ref. 29), which indicates a quick formation of dynamically relaxed systems and their possible notable role in stellar migration to the nuclear region. Using Atacama Large Millimeter/submillimeter Array (ALMA) observations, previous works reported the existence of cold gaseous disks at $z ≈ 5$ (refs. 10,11). However, we observationally confirm that also stellar disks could be dynamically cold at these early times.

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

## Methods

### Cosmological model

We assume a flat ΛCDM cosmology with Hubble constant $H_0 = 67.7 \, \mathrm{km \, s^{-1} \, Mpc^{-1}}$ and matter density $\Omega_m = 0.310$ (ref. 34). All magnitudes are in the absolute bolometric system.

### Data

JWST/NIRCam data used in this work were taken during the first epoch (21–22 June 2022) of the CEERS program, one of 13 early release science surveys approved for JWST Cycle 1. In particular, we focus on data from CEERS pointing labelled NIRCam1, which is covered with seven filters: F115W, F150W, F200W, F277W, F356W, F410M and F444W (ref. 35). The final mosaics in all of the filters have a pixel scale of 0.03 arcsec px$^{-1}$ and a PSF full-width at half-maximum (FWHM) ranging from 0.066 to 0.161 arcsec, reaching a point-source limiting magnitude (5$\sigma$) of approximately 29 mag (refs. 35,36). The root mean square astrometric alignment quality is less than approximately 5–10 mas per source between NIRCam filters[35]. In Extended Data Fig. 1, we show the cutouts of ceers-2112 in all NIRCam bands.

For studying the morphology of ceers-2112, we built a stack image combining all seven NIRCam bands. We converted individual images in counts and PSF convolved them to match the angular resolution of the F444W image. Empirical PSFs for the CEERS datasets are created as described in ref. 36, whereas kernels to match bluer PSFs to F444W are created using the pypher Python-based routine[37]. Finally, we combined all PSF-convolved images using the ccdproc.combine v.2.4.0 astropy image reduction package[38].

For studying the stellar population properties of ceers-2112, we extended the NIRCam wavelength baseline with HST images (F606W, F814W, F125W, F140W and F160W) from the Cosmic Assembly Near-infrared Deep Extragalactic Legacy Survey (CANDELS) collaboration[39,40]. These data were recalibrated by the CEERS team and drizzled to match the same angular scale of the JWST observations[35] (v.1.9).

### Morphology

We analysed the morphology of ceers-2112 by modelling its surface-brightness distribution to characterize its structural components using four different diagnostics: (1) isophotal analysis; (2) Fourier decomposition; (3) one-component Sérsic photometric modelling; and (4) two-component bar + disk photometric modelling.

Firstly, we considered the radial surface-brightness profile of ceers-2112 and modelled its isophotes with the photutils.isophote astropy package[41,42] using three short wavelength bands (that is, F115W, F150W and F200W; Extended Data Fig. 2a,d) and three long wavelength bands (that is, F277W, F356W and F444W; Extended Data Fig. 2b,e). We created ellipticity and position angle profiles by keeping the centre fixed to the average value measured in the inner region of the galaxy. Then, we checked that ceers-2112 satisfied the criteria[19,43,44] of hosting a putative bar-dominated region: (1) the galaxy became elongated in the bar region (ellipticity $\varepsilon > 0.25$) and the position angle remained almost constant along the bar (|Δ position angle| < 15°); and (2) the ellipticity dropped in the outer region of the galaxy (Δ$\varepsilon$ = 0.1), where the disk component dominates. Our findings suggest the presence of a bar, which appears more prominent at longer wavelengths, with an ellipticity always greater than 0.4 up to a radius of $r \approx 0.45$ arcsec. It is worth noting that the analysis of individual bands is complicated by the presence of spiral arms, which could drive the mild change in position angle (and slightly affect the ellipticity) in the outskirts of the galaxy. As a caveat, the disk component is very mildly detected (in particular, in short wavelength bands), leading to a small change in bar-to-disk $\varepsilon$ and position angle. For this reason, we decided to further analyse the morphology of ceers-2112 using the combined image obtained by stacking all NIRCam filters, to increase the final signal-to-noise ratio, in particular, in the outskirts of the galaxy. In the combined image (Extended Data Fig. 2c,f), our analysis showed an inner bar-dominated region ($\varepsilon > 0.4$, Δ position angle < 15°), a region where mild spiral arms develop from the barred structure and the outer disk-dominated region, where the ellipticity and position angle drop[45,46].

Secondly, we analysed the deprojected combined stack image of ceers-2112 and decomposed its azimuthal luminosity surface-density distribution into the Fourier $m$-components[47]. To project the galaxy into the face-on view keeping the flux preserved, the image was stretched along the disk minor axis by a factor of $\cos(i_{disk})^{-1}$, where $i_{disk}$ is the disk inclination derived from the disk ellipticity. In particular, from the isophotal fitting of the combined image we derived $\varepsilon_{disk} = 0.23$ ($i_{disk} = 41°$), taking the median values in the outer isophotes (0.6 < $r$ < 0.7 arcsec) where the influence of the bar is negligible (Extended Data Fig. 2c,f). In Fig. 1d, we show the radial profiles of the relative amplitude of the $m = (2, 4, 6)$ components. In particular, the $m = 2$ component shows the characteristic behaviour of bars[14,16]: increasing with radius (with a prominent peak $I_2/I_0 > 0.4$) and then decreasing in the disk region. The phase angle $\phi_2$ of the $m = 2$ component is quite constant in the bar region (|Δ$\phi_2$| < 10° with respect to the $I_2/I_0$ peak), which provides an additional confirmation of the presence of the bar component. We further tested our findings by repeating the Fourier decomposition assuming both different position angles (Δ position angle ± 5°) and inclinations (Δ$i$ ± 5°) for the galaxy (eight different configurations). No systematics were found in the bar identification due to galaxy deprojection effects. Furthermore, it is worth reporting that the bar/interbar intensity contrast based on the Fourier decomposition provides results about the length and strength of the bar that are consistent with those of the $m = 2$ Fourier analysis[14,48]. It is also worth noting that our Fourier analysis allows us to rule out the possibility that the stellar bar could be misled by spiral arms developing from a compact bulge. Indeed, this latter case would not produce an $m = 2$ peak in the inner region of the galaxy[16].

Thirdly, to disentangle the contribution to the surface brightness of bar and spiral arms, we modelled the galaxy with a single Sérsic component and looked at the residual image (Fig. 1b). We used the Python package statmorph[49] to retrieve both the parametric and non-parametric morphology of the galaxy. The best-fitting model provides a quite low Sérsic index $n = 0.65$ (disky galaxy), with the residual image highlighting prominent features in correspondence of the spiral arms and edges of the bar component. Our findings suggest that the one-component Sérsic model is not sufficient to describe the complex morphology of ceers-2112.

Finally, we perform a 2D photometric decomposition of ceers-2112 using the galaxy surface photometry 2D decomposition algorithm (GASP2D[50,51]). We model the galaxy (Fig. 2c) by assuming that its surface-brightness distribution is the sum of a double-exponential disk[52] and a Ferrers bar[53]. GASP2D returns the best-fitting values of the structural parameters of each morphological component by minimizing the $\chi^2$ after weighting the surface brightness of the image pixels according to the variance of the total observed photon counts due to the contribution of both galaxy and sky (Extended Data Fig. 4c,d). Because GASP2D does not fit the spiral arm components, we mask them to avoid possible contamination in retrieving the ellipticity and position angle of the bar. The mask for the 2D bar + disk decomposition is built by growing the spiral arms residuals, excluding the bar region. Because the formal errors obtained from the $\chi^2$ minimization are usually not representative of the real errors, we estimated the uncertainties on the bar and disk parameters by analysing a sample of images of mock galaxies built with Monte Carlo simulations[54].

As a caveat, as the composite stack image covers the wavelength range from the rest-frame ultraviolet to near infrared, dust attenuation and spatially variable younger stellar populations may result in a composite light distribution that does not follow the stellar distribution. To support our analysis, we then created combined short wavelength (F115W, F150W and F200W) and long wavelength (F277W,

F356W and F444W) stack images, following the procedure described in the previous section. The short wavelength stack image was convolved to F200W, whereas the long wavelength stack image was convolved to F444W. The isophotal analysis of short wavelength stack and long wavelength stack images is shown in Extended Data Fig. 3c,d). We see a similar trend at short and long wavelengths, with two main differences: (1) the position angle is almost constant in the long wavelength stack image, although it shows a mild variation in the short wavelength stack image, making the identification of the bar less clear in the ultraviolet–optical rest-frame regime; and (2) the signal-to-noise ratio, in particular, in the inner and outer regions, is very low in the short wavelength stack image with respect to the long wavelength stack image. The Fourier analysis of these images is shown in Extended Data Fig. 3e. We see that the bar component is clearly detected at longer wavelengths ($m = 2$ component stronger than any other component), whereas, at shorter wavelengths, we see both prominent $m = 1$ and $m = 2$ components. This is due to the non-asymmetry (lopsidedness) of the elongated structure seen at short wavelengths. Again, while the evidence for the bar structure is present both at short and long wavelengths, the bar is more evident in the redder bands, as expected from near infrared studies in the local Universe[55].

For the reasons described above, we based our main analysis on the image obtained by combining all seven NIRCam bands. Our morphological analysis provided four independent estimations of the bar length: (1) $R_{bar,1} = 0.49 \pm 0.09$ arcsec, from the outer radius of the FWHM of $I_2/I_0$ (ref. 56); (2) $R_{bar,2} = 0.44 \pm 0.04$ arcsec, from the outer radius of the FWHM of the bar/interbar contrast[48]; (3) $R_{bar,3} = 0.49 \pm 0.02$ arcsec, from the radius at which there is the first minimum after the (deprojected) ellipticity peak[45]; and (4) $R_{bar,4} = 0.42 \pm 0.03$ arcsec, from the Ferrers bar modelling[53].

### Redshift estimation
We carefully measured the ACS, WFC3 and NIRCam photometry with the rainbow code[57,58], using small elliptical apertures (radius of 0.44 arcsec; $\varepsilon = 0.35$) to retrieve reliable colours and avoid possible photometric contamination by a foreground extended source ($z_{phot} = 1.1$; projected distance of approximately 3.5 arcsec). Then, we measured the photometry on slightly larger apertures (radius of 0.84 arcsec) to obtain the integrated emission. Finally, we normalized the SED measured on small apertures using the median difference of the flux measured in the small and large apertures. Photometric errors were estimated by measuring the background noise locally around ceers-2112, which accounted for correlated noise introduced by drizzling the ACS, WFC3 and NIRCam images[57,59].

The fiducial photometric redshift was derived using EAZYpy[60] including the tweak_fsps_QSF_12_v.3 set of 12 Flexible Stellar Population Synthesis templates[61,62]. The combined HST + JWST SED, the values of the photometric redshift and the corresponding probability density functions are shown in Extended Data Fig. 5. We further tested the photometric redshift estimation against different codes (that is, Dense Basis[63]; Prospector[64]) and found consistent results (see Extended Data Fig. 5, inset panel).

### Stellar population properties
We derived the fiducial spatially resolved SFH of ceers-2112 with synthesizer[57], assuming delayed-exponential SFHs. We adopt timescale values $\tau$ between 100 Myr and 5 Gyr, ages between 1 Myr and the age of the Universe at the redshift of ceers-2112, the entire set of discrete metallicities provided by the Bruzual and Charlot models[65], a Calzetti et al. attenuation law[66] with $V$-band extinction values between 0 and 5 mag and a Chabrier initial mass function[67]. The nebular continuum and emission lines were added to the models[57].

We further tested the systematics related to the stellar population modelling by deriving the integrated stellar population properties of ceers-2112, using both parametric and non-parametric SFHs (Extended Data Fig. 6 and Extended Data Table 1). For this purpose, we fitted the integrated HST + NIRCam photometry using the Fitting and Assessment of Synthetic Templates (FAST) code[68], Dense Basis[63] and Prospector[64]. For the FAST algorithm, we assumed an exponentially declining star formation history. We used Bruzual and Charlot stellar population synthesis models[65], Calzetti et al. extinction law[66] with attenuation $0 < A_V < 4$ mag, and a Chabrier initial mass function[67]. For Dense Basis, we use a uniform prior for the stellar mass $\log(M_\star/M_\odot)$ between 7 and 12, uniform prior for the metallicity $\log(Z/Z_\odot)$ between $-1.5$ and 0.25, a Calzetti et al. attenuation law[66] with exponential prior and $V$-band extinction values between 0 and 4 mag, and a Chabrier initial mass function[67]. For Prospector[64,69], we used both a delayed-exponential and a non-parametric SFH. For the $\tau$-model, we used stellar ages ranging between 1 Myr and the age of the Universe at the redshift of ceers-2112 and the star formation scale in the range $0.1 < \tau < 20$ Gyr. For the non-parametric model, we used an SFH with the continuity prior[69]. We adopted five lookback time bins in this fit, with the star formation rate being constant within each bin. The first bin was fixed at $0 < t < 30$ Myr to capture the recent episodes of star formations. We used uniform priors on all of the following parameters: stellar mass $\log(M_\star/M_\odot)$ between 5 and 12, metallicity $\log(Z/Z_\odot)$ between $-1.5$ and 0.5 and effective $V$-band optical depth between 0 and 5 mag. We adopted the Chabrier initial mass function[67] and the Calzetti et al. dust attenuation law[66].

### Data availability
This study used CEERS JWST/NIRCam data, which are publicly available from the Mikulski Archive for Space Telescopes (MAST; http://archive.stsci.edu) under program ID 1345 (principal investigator: Finkelstein). Calibrated NIRCam data products from the CEERS team are available at https://ceers.github.io/releases.html.

### Code availability
JWST NIRCam data are calibrated using the JWST Pipeline[70] (v.1.7.2, reference mapping 0989; https://github.com/spacetelescope/jwst). Photometric redshifts and/or stellar population properties are measured using EAZYpy[60], FAST[68], synthesizer[57], Dense Basis[63] and Prospector[64]. The morphological analysis was performed using photutils.isophote[41,42], statmorph[49] and GASP2D[50]. The Fourier analysis is based on the implementation described in ref. 14.

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

**Acknowledgements** We thank S. Roca-Fàbrega and E. Borsato for their comments. L.C. acknowledges financial support from the Comunidad de Madrid under Atracción de Talento grant no. 2018-T2/TIC-11612. L.C. and P.G.P.-G. acknowledge support from grant nos. PGC2018-093499-B-I00, PID2022-139567NB-I00 and MDM-2017-0737 Unidad de Excelencia 'Maria de Maeztu' Centro de Astrobiología (INTA-CSIC) funded by the Spanish Ministry of Science and Innovation/State Agency of Research MCIN/AEI/ 10.13039/501100011033, FEDER, UE. C.C. acknowledges support from grant nos. PRE2019-087503 and PID2021-123417OB-I00 funded by MCIN/AEI/ 10.13039/501100011033 'ESF Investing in your future' and 'ERDF A way of making Europe', respectively. J.M.A. acknowledges the support of the Viera y Clavijo Senior program funded by ACIISI and ULL and the support of the Agencia Estatal de Investigación del Ministerio de Ciencia e Innovación (MCIN/AEI/10.13039/501100011033) under grant nos. PID2021-128131NB-I00 and CNS2022-135482 and the European Regional Development Fund (ERDF) 'A way of making Europe' and the 'NextGenerationEU/PRTR'. F.B. and J.V.F. acknowledge the support from grant no. PID2020-116188GA-I00 by the Spanish Ministry of Science and Innovation, and F.B. also acknowledges grant no. PID2019-107427GB-C32. Support for this work was provided by NASA through grant no. JWST-ERS-01345 awarded by the Space Telescope Science Institute, which is operated by the Association of Universities for Research in Astronomy, Inc., under NASA contract NAS 5-26555.

**Author contributions** L.C. discovered the galaxy presented in this manuscript and led the analysis of the data and the preparation of the manuscript. P.A.H., M.B.B., M.D., S.L.F., J.S.K., A.M.K., C.P. and N.P. led the NIRCam/JWST observing proposal. M.B.B. and A.M.K. led the calibration of the JWST/NIRCam, HST/ACS and HST/WFC3 datasets. P.G.P.-G., C.C. and S.L.F. contributed to preparing the empirical PSF and estimating accurate photometric errors. Y.G., C.B., S.J., J.S.K., C.C., J.M.A., E.M.C. and L.M. contributed to measuring the structural parameters of the bar. P.G.P.-G., G.B., A.V., K.G.I., L.B. and Y.C. contributed to the redshift estimation and stellar population analysis. L.C., P.G.P.-G., Y.G., C.B., S.J., C.C., E.M.C., J.M.A., F.B., M.C.C., A.D., M.G., B.W.H., M.H.C., R.A.L., L.M.S., J.V.F., S.W. and L.Y.A.Y. contributed substantially to the discussion, analysis and interpretation of the results. All authors assisted with the preparation of the final published manuscript.

**Competing interests** The authors declare no competing interests.

**Additional information**
**Correspondence and requests for materials** should be addressed to Luca Costantin.

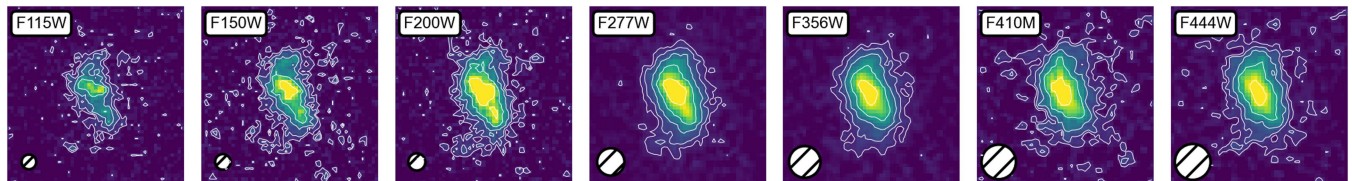

**Extended Data Fig. 1 | Multiwavelength view of ceers-2112.** Postage stamps of ceers-2112 (RA = 214.97993 degrees; DEC = 52.991946 degrees; J2000.0) in all NIRCam filters used in this work. The cutouts are 53 × 53 px², which corresponds to 1.59 × 1.59 arcsec² (12.5 × 12.5 kpc² at $z$ = 3.03). We report the angular resolution as 2 × FHWM of the PSF and the isophotal contours (white solid lines).

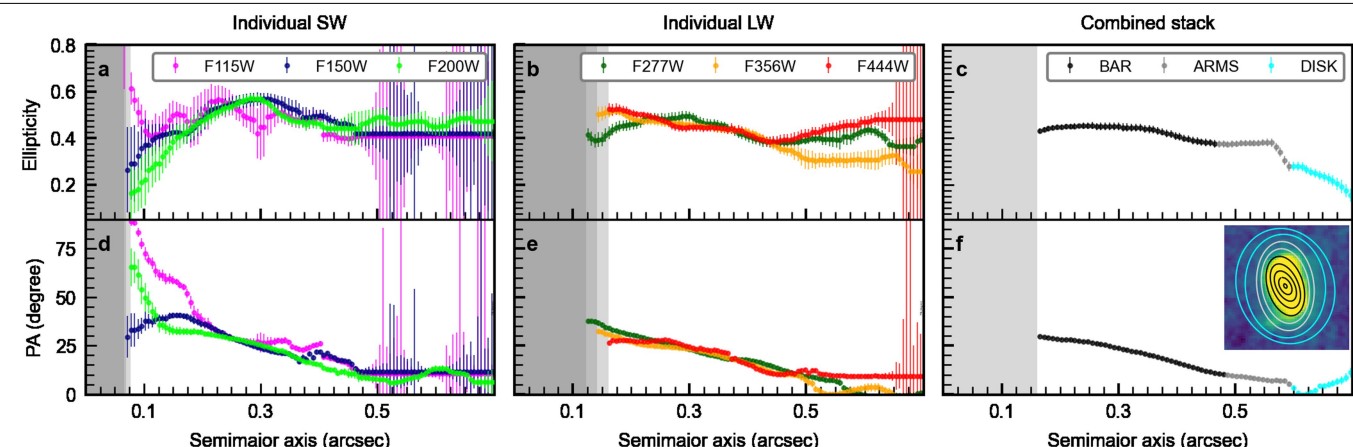

**Extended Data Fig. 2 | Isophotal analysis of ceers-2112.** In each panel, the shaded regions mark the size of the PSF FWHM in the different bands, while error bars show the 1σ standard deviation of each point. **a**, Radial profiles of ellipticity derived from the isophotal analysis of ceers-2112 in the F115W (pink), F150W (blue) and F200W band (light green). **b**, Radial profiles of ellipticity derived from the isophotal analysis of ceers-2112 in the F277W band (dark green), F356W (orange) and F444W band (red). **c**, Radial profiles of ellipticity derived from the isophotal analysis of ceers-2112 in the combined stack image (all seven NIRCam filters). The region of the bar, spiral arms and outer disk are shown as black, grey and cyan datapoints. **d**, As panel **a**, but for the position angles. **e**, As panel **b**, but for the position angles. **f**, As panel **c**, but for the position angles. The inset panel shows some of the ellipses superposed to the composed stack image (1.59 × 1.59 arcsec²).

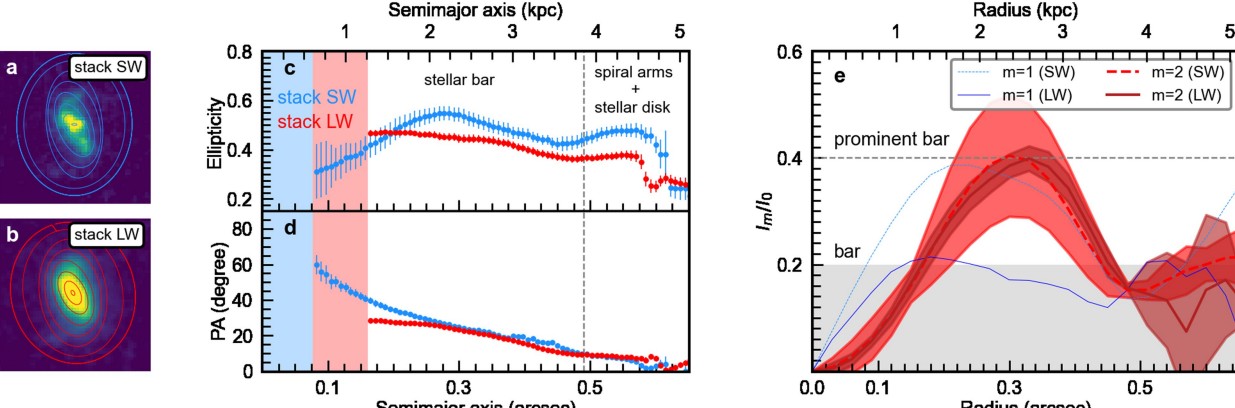

**Extended Data Fig. 3 | Isophotal and Fourier analysis of SW and LW stack images of ceers-2112. a**, Postage stamp of the stack SW image (F115W, F150W and F200W) with some of the ellipses superposed (1.59 × 1.59 arcsec²). **b**, Postage stamp of the stack LW image (F277W, F356W and F444W) with some of the ellipses superposed (1.59 × 1.59 arcsec²). **c**, Radial profiles of ellipticity derived from the isophotal analysis of ceers-2112 in the stack SW image (blue) and stack LW image (red). The shaded regions mark the size of the PSF FWHM in the different bands, while error bars show the 1σ standard deviation of each point. **d**, As **c**, but for the position angles. **e**, Radial profiles of the relative amplitude of the $m = 1$ (blue lines) and $m = 2$ (red lines; shaded regions: 1σ confidence intervals) Fourier components derived from the deprojected combined SW image (dashed lines) and the deprojected combined LW image (solid lines) of ceers-2112.

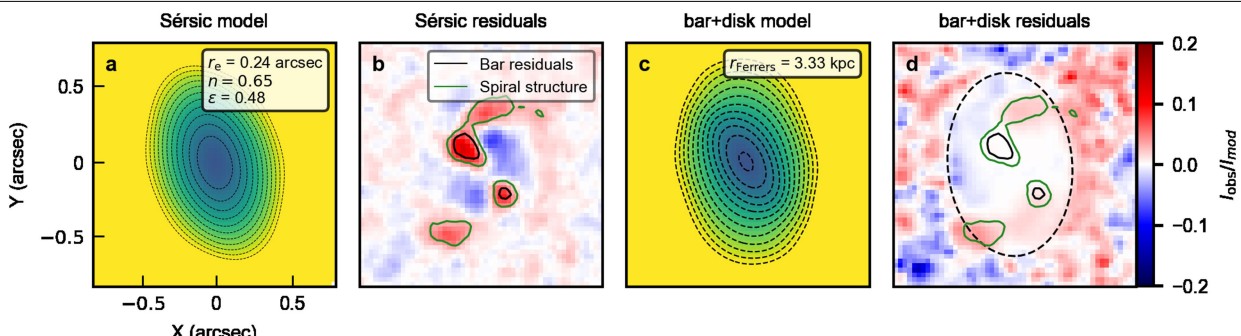

**Extended Data Fig. 4 | Parametric morphological modeling of ceers-2112. a**, One-component Sérsic model of ceers-2112. **b**, One-component Sérsic residuals, which highlight the bar and spiral structures (black and green contours, respectively) **c**, Two-dimensional bar+disk model, which shows a stellar bar of length $r_{Ferrers} = 0.42 \pm 0.03$ arcsec (3.3 kpc). **d**, Two-dimensional

bar+disk model residuals. The bar and spiral structures (black and green contours, respectively) are superposed to the image. The black dashed line marks the break radius of the double-exponential disk model, where the surface brightness of the model rapidly declines.

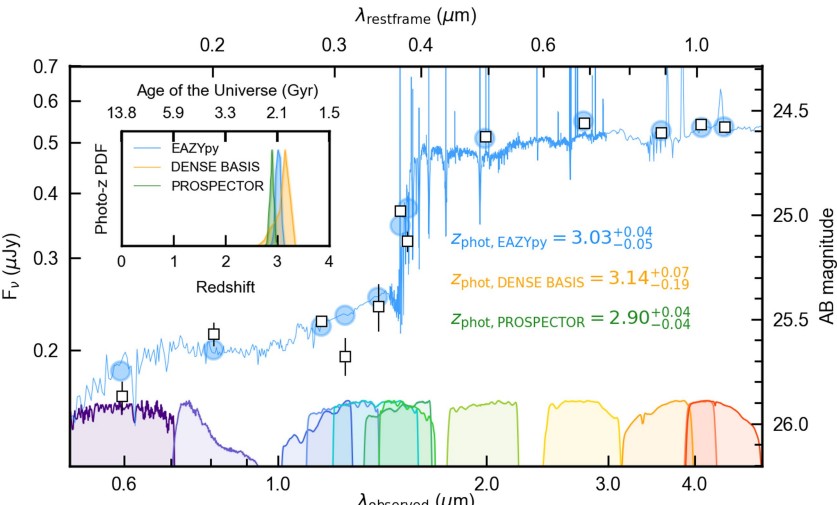

**Extended Data Fig. 5 | SED and redshift of ceers-2112.** Black empty squares (blue circles) denote our fiducial (model) photometry from HST/ACS + WFC3 and JWST/NIRCam instruments, respectively. The EAZYpy model spectrum is shown in blue. Error bars show the 1σ standard deviation of each point. The inset plot shows the P(z) distributions.

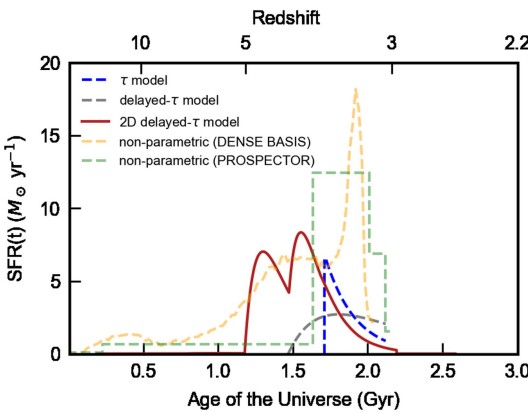

**Extended Data Fig. 6 | SFH modeling of ceers-2112.** Comparison of different model assumptions: exponentially-declining SFH (τ-model; FAST code; blue dashed line); delayed exponentially-declining SFH (delayed-τ model; Prospector; gray dashed line); two-dimensional delayed exponentially-declining SFH (2D delayed-τ model; synthesizer; red solid line); non-parametric SFH (Dense Basis and Prospector; orange and green dashed lines, respectively).

**Extended Data Table 1 | Mass and *SFR*$_{50}$ of ceers-2112 derived from different model assumptions**

|  | 2D delayed-$\tau$ | delayed-$\tau$ | $\tau$-model | non-parametric | non-parametric |
|---|---|---|---|---|---|
| code | synthesizer | PROSPECTOR | FAST | PROSPECTOR | DENSE BASIS |
| $M_\star$ $(M_\odot)$ | $3.9 \times 10^9$ | $4.9 \times 10^9$ | $2.6 \times 10^9$ | $4.5 \times 10^9$ | $4.6 \times 10^9$ |
| $\log(SFR_{50})$ $(M_\odot \ \mathrm{yr}^{-1})$ | $-0.54$ | $0.33$ | $-0.01$ | $0.73$ | $0.31$ |