## [Peer Review File · Nature]

Manuscript Title: A Milky-Way-like barred spiral galaxy at a redshift of 3

Reviewer Comments & Author Rebuttals

Reviewer Reports on the Initial Version:

Referees' comments:

Referee #1 (Remarks to the Author):

Referee report on paper submitted to Nature on:

``The eldest sibling of the Milky Way: JWST reveals a barred galaxy at $z\sim 3$ ''

by anonymous authors.

The authors present a morphological analysis of the JWST NIRCам images of CEERS-2112 at $z=3$, which is a galaxy with stellar mass $M = 2.6 \times 10^9 M_{\odot}$ that may already have developed a central bar in the first 2 Gyrs. Bars are thought to develop in dynamically cold stellar disks at low redshifts ($z < 2$), while gas turbulence in higher redshift ($z > 3$) disk galaxies is thought to suppress bar formation. The implication of this finding would be that that high-redshift bars could form within 400-700 Myr, so that at least some dynamically cold stellar disks would already be in place at $z > 4-5$, i.e., more than 12 Gyrs ago. This would be a major discovery, if proven, that could be worthy publication in Nature. I do have a number of reservations that the authors will need to address before the paper can be considered for publication:

1) The image convolution to the same PSF size and structure: The authors state on line 121 that they did this, but details need to be given on exactly how this was done. Most PSF-matching techniques are accurate for integrated photometry, but not necessarily so when measuring resolved structure in faint galaxies at various spatial scales. Hence, to judge how well the bar/disk/spiral-arm structure can be delineated as a function of wavelength, the authors need to describe the technique used for PSF matching in more detail, and how this may have affected lower surface brightness structures as a function of wavelength.

2) The authors state a number of times that they perform the bar/disk/spiral arm decomposition and Fourier analysis in Fig. 2--3 on the NIRCам F200W images, as well as on the stack of the NIRCам images in all 7 CEERS filters. This is fine, but this analysis overlooks one powerful aspect of bar studies: The current main F200W filter samples the bar at a restframe wavelength of $\sim 5000 \text{ \AA}$ at $z=3$, where bars are known not to be very strong. At lower redshifts, bars are known to increase significantly in strength at the longer restframe wavelengths (1--2 μm ; refer here, e.g., to the work of Eskridge et al. 2000, AJ, 119, 536). This then means that the claimed bar as contrasted to the disk at $z\sim 3$ --- after properly performing the PSF-matching in 1) --- should stand out a lot better in the F277W, F356W and F444W filters, perhaps also in the lower S/N F410M filter.

This should be addressed in detail, as it could form stronger evidence in favor of the author's argument. As it stands, I am not convinced that Fig. 3 doesn't

just show a central (spheroidal) bulge instead of a real bar. But the above suggestion could certainly convince us otherwise. And while this analysis is done for the longer NIRCcam wavelengths separately, it should also be done for the HST blue filters separately (perhaps after stacking these also) to more convincingly show that the bar of this object increases in strength towards the longer rest-frame wavelengths.

3) Given the above reservations, some of the statements on the paper are currently overstated or not proven. E.g., search for "robust" and "the first", and these need to be reworded more conservatively.

Other comments:

Fig. 1: The central wavelength of the 0.8 μm data point certainly, and perhaps also those of some of the 1.1-1.6 μm data points do not seem to match the filter transmission curves plotted.

Referee #2 (Remarks to the Author):

This manuscript presents the discovery of a stellar bar structure in a Milky Way progenitor mass ($\sim 10^9 M_{\text{sun}}$) galaxy at $z \sim 3$. This result of a stellar bar structure in a lower mass galaxy within the first 2 Gyr of the Universe is novel and timely, providing important insights into both the properties of high redshift galaxies (that stellar disks can be dynamically cool and form bars) and the formation and evolution of galaxies over time, including our own Milky Way (as stellar bars can help to funnel gas inwards which is necessary for the growth of bulge structures and central super massive black holes).

While the topic of study is novel and timely, some of the analysis does not robustly and unambiguously support the specific stated conclusions. With revision including extended analysis or explicit presentation of existing alternative analysis, this manuscript certainly merits reconsideration for publication.

Major concerns:

1. SFHs: (Sections 3.1.2 & 3.1.3):

The inferred SFH (both integrated and resolved) for ceers-2112 is a critical component of the conclusion that this galaxy rapidly formed a bar in < 400 Myr.

However, it is not clear that the conclusion of rapid bar formation is robust.

The fiducial integrated fit (using FAST) is performed using an exponentially declining SFH, which may or may not be appropriate for this system and does place prior constraints on the measured SFH (eg, Carnall et al. 2019, doi:10.3847/1538-4357/ab04a2). Why was an exponentially declining SFH adopted, compared to a delayed exponentially declining SFH as with the Synthesizer fitting?

Moreover, the full set of fits performed (including both parametric and non-parametric SFHs) exhibit a wide range of star formation timescales. These uncertainties in the recovered SFH imply that the bar formation timescale is also uncertain.

Nonetheless, even if the bar formation timescale is more extended or uncertain (from $< \sim 400$ Myr - 1 Gyr), the early epoch and low stellar mass of this galaxy still makes the bar structure remarkable.

2. Bar structure measurements: (Section 3.2):

(a) Sections 3.2.2-3.2.4:

The details of the stacked images used for the Fourier analysis and morphological modeling is never stated. Presumably all available JWST/NIRCam bands are used in the stack (F115W, F150W, F200W, F277W, F356W, F410M, F444W), but this is not explicitly stated and should be clarified. If this is the case, this stack covers from the rest-UV to rest-NIR, and thus dust attenuation and spatially-variable younger stellar populations may result in a composite light distribution that does not follow the full stellar distribution. The authors are certainly aware of this issue, and likely used the stack because of signal-to-noise considerations required for the detailed structural analysis. However, if at all possible given the imaging depth, it would be interesting to see the detailed structural analysis performed on stack of only the redder bands (F277W and redder), which would mitigate many of the concerns above by probing only the redder portions of the rest optical and NIR while also increasing the composite image depth. If this is not possible, the potential impact of such factors should be explicitly discussed in the text.

(b) Section 3.2.1: While F200W does probe the rest frame optical at $z \sim 3$ and provides very high spatial resolution, the isophotal analysis could be strengthened by also performing isophotal fits on the F444W band, which probes the stellar continuum and thus provides a much more direct constraint on the stellar distribution in this galaxy.

Minor comments:

Abstract, line 17, and elsewhere in the text:

This work shows dynamically cold stellar disks for *some* galaxies can be in place *by* $z=4-5$ (depending on the exact inferred SFH), but does not address earlier epochs, nor does this analysis address the universality of cold stellar disks at this epoch and stellar mass.

Data / Sec 2:

Lines 72-75:

It appears the publicly-available reduced NIRCam mosaics from the CEERS team were used for this analysis given the references, but this should be explicitly stated.

Line 76:

Please list the HST filters which were included.

Figures:

- Figs 1 & 2: Image postage stamps and resolved property maps

No size bars have been included, which makes it difficult for the reader to evaluate the physical scale of the galaxy and how the various photometric apertures and structural analysis profiles compare to the observed galaxy extent.

- Fig 1: The contours in the multiwavelength postage stamps should be defined

Results / Sec 3:

Lines 81-83:

How was this galaxy identified as barred for this analysis, given it was not labeled as barred in the visual inspection from Kartaltepe et al. 2023?

Sec 3.1, lines 86-90:

What is the ellipticity of the small photometric apertures, and how was this determined?

How much of the galaxy's structure is captured within these small apertures?

What are the radii and ellipticity of the large apertures?

The normalization procedure is ambiguous as phrased: was the integrated photometry measured on a single band and then the same correction factor applied to all the 0.44" radius color aperture

photometry?

Exactly which set of photometric measurements were used in the integrated SED fits: a set of full-band larger aperture photometry, the normalized smaller aperture photometry, or some other case?

Is the nearby extended source a foreground object?

Line 101:

"... show that ceers-2112 accreted the bulk of its stars ..." Formed the bulk of its stars? Or is there evidence for interactions through which stars could have been accreted?

Lines 101-102:

These timescale constraints are not necessarily universal, even for this epoch and galaxy stellar mass.

Lines 102-104:

How do the stellar masses and SFRs from the alternative SED fits compare to the quoted values from FAST?

Sec 3.2:

As before, the filters included in the stacked image should be stated.

Sec 3.2.2:

How was the deprojection for the Fourier analysis performed? How was the inclination of the system determined?

Sec 3.2.3 & Fig 3, bottom center panel:

How were the residual contours ascribed to the spiral versus bar structures?

Sec 3.2.4 & Fig 3, bottom fourth and fifth panels:

What is masked in these panels and the two-component morphological model fitting? How is that mask determined?

Sec 3.2.4, lines 166-167:

How does the two-component fit Ferrers bar length compare to the length inferred from the Fourier analysis?

Sec 4, lines 203-205, and Sec 3.1.3:

For the non-expert reader, it would be helpful to explicitly lay out the logic that rapid gas consumption, as inferred from the SFH, allows for the stellar disk to become dynamically cool and form a bar within a short time afterwards.

We are grateful to the referees for a careful review and for the valuable comments that we have made our best to take into account. To facilitate the reading of the report we have formatted the referee's comments into sections and enumerated points below. Our response is given in bold following each point.

The manuscript is in a substantially different shape with respect to the original submitted version since we took into account both the editorial changes needed to present it in a style suited for Nature journal and the suggestions from the different referees.

Referee #1 (Remarks to the Author):

The authors present a morphological analysis of the JWST NIRCам images of CEERS-2112 at $z=3$, which is a galaxy with stellar mass $M = 2.6 \times 10^9 M_{\odot}$ that may already have developed a central bar in the first 2 Gyrs. Bars are thought to develop in dynamically cold stellar disks at low redshifts ($z < 2$), while gas turbulence in higher redshift ($z > 3$) disk galaxies is thought to suppress bar formation. The implication of this finding would be that that high-redshift bars could form within 400-700 Myr, so that at least some dynamically cold stellar disks would already be in place at $z > 4-5$, i.e., more than 12 Gyrs ago. This would be a major discovery, if proven, that could be worthy publication in Nature. I do have a number of reservations that the authors will need to address before the paper can be considered for publication:

1) The image convolution to the same PSF size and structure: The authors state on line 121 that they did this, but details need to be given on exactly how this was done. Most PSF-matching techniques are accurate for integrated photometry, but not necessarily so when measuring resolved structure in faint galaxies at various spatial scales. Hence, to judge how well the bar/disk/spiral-arm structure can be delineated as a function of wavelength, the authors need to describe the technique used for PSF matching in more detail, and how this may have affected lower surface brightness structures as a function of wavelength.

We agree with the referee that the details about the PSF-matching were missing in the manuscript. The full description of the selection of stars and the procedure to create the PSFs and the matching kernels are described in Finkelstein et al. 2023, Sect. 3.2 and Table 1 (following the prescription in Finkelstein et al. 2022).

To address this matter, we add the following sentence in Methods (Data, lines 202-208): "For studying the morphology of ceers-2112, we build a stack image combining all 7 NIRCам bands. We convert individual images in ADU and PSF-convolve them to match the angular resolution of the F444W image. Empirical PSFs for the CEERS datasets are created as described in [36], while kernels to match bluer PSFs to F444W are created using the pypher Python-based routine [37]. Finally, we combine all PSF-convolved images using the ccdproc.combine v2.4.0 astropy image reduction package [38]."

We complement the analysis based on the stack image with a more careful isophotal and Fourier analysis at different wavelengths (see #2).

2) The authors state a number of times that they perform the bar/disk/spiral arm decomposition and Fourier analysis in Fig. 2--3 on the NIRCам F200W images, as well as on the stack of the NIRCам images in all 7 CEERS filters. This is fine, but this analysis overlooks one powerful aspect of bar studies: The current main F200W filter samples the bar at a restframe wavelength of $\sim 5000 \text{ \AA}$ at $z=3$, where bars are known not to be very strong. At lower redshifts, bars are known to increase significantly in strength at the longer restframe wavelengths (1--2 μm ; refer here, e.g., to the work of Eskridge et al. 2000, AJ, 119, 536). This then means that the claimed bar as contrasted to the disk at $z\sim 3$ --- after properly performing the PSF-matching in 1) --- should stand out a lot better in the F277W, F356W and F444W filters, perhaps also in the lower S/N F410M filter.

This should be addressed in detail, as it could form stronger evidence in favor of the author's argument. As it stands, I am not convinced that Fig. 3 doesn't just show a central (spheroidal) bulge instead of a real bar. But the above suggestion could certainly convince us otherwise. And while this analysis is done for the longer NIRCcam wavelengths separately, it should also be done for the HST blue filters separately (perhaps after stacking these also) to more convincingly show that the bar of this object increases in strength towards the longer rest-frame wavelengths.

We thank the referee for allowing us to clarify this point. Indeed, the barred structure of ceers-2112 is clearly identified by analyzing the restframe NIR images. We rephrase the text to clarify that the main morphological analysis (i.e., Fourier decomposition, one-component modeling, and bar+disk modeling) is performed on the combined image (“SW+LW”: stacking all seven bands). We also add the isophotal analysis of the combined stack image (SW+LW) in the new Extended Figure 5, and comment on it in Methods (Morphology, lines 237-243).

We repeat the isophotal analysis on six individual bands, three SW (F115W, F150W, F200W) and three LW (F277W, F356W, F444W). The results are shown in the new Extended Data Figure 5. We see that an elongated structure (ellipticity > 0.4) is clearly evident in the LW bands, while the signature for the bar is less clear (but still present) at shorter wavelengths. In particular, in the SW images the stellar disk is not detected up to F200W.

Furthermore, following the referee’s suggestion, we create two new combined images, one stacking three SW bands (F115W, F150W, F200W; convolved both to F200W and to F444W) and one stacking three LW bands (F277W, F356W, F444W; convolved to F444W). We add to this report the isophotal analysis of “stack SW” and “stack LW” images (Figure A; see also new Extended Data Figure 6). We see a similar trend at short and long wavelengths, with two main differences: (1) the position angle is almost constant in the stack LW image, while it presents more variation in the “stack SW” image (especially the one convolved to F200W), making the identification of the bar less clear in the UV-optical restframe regime; (2) the S/N, especially in the inner and outer regions, is very low in the “stack SW” image with respect to the “stack LW” image. This is now mentioned in Method (Morphology, lines 298-304).

Figure A. (from left to right) Isophotal analysis of ceers-2112 on the stack SW (convolved to F200W), on the stack SW (convolved to F444W), and on the stack LW (convolved to F444W).

Finally, we repeat the Fourier analysis of the “stack SW” and “stack LW” images (see Methods, Morphology, lines 304-311). The results are shown in the new Extended Figure 6. We see that the bar component is clearly detected in the “stack LW” ($m=2$ even component stronger than any other component), while the “stack SW” image shows both prominent $m=1$ and $m=2$ components. This is due to the non-asymmetry (lopsidedness) of the elongated structure seen at short wavelengths. Again, while the evidence for the bar structure is present both at short and long wavelengths, the bar is more evident in the “stack LW”. The bar length derived from the Fourier analysis and the isophotal fitting in the “stack SW” and “stack SW+LW” are totally compatible (see Table A hereafter). For these reasons, we base our main analysis on the “stack SW+LW” image.

	$R_{\text{bar},1}$ (arcsec)	$R_{\text{bar},2}$ (arcsec)	$R_{\text{bar},3}$ (arcsec)
LW	0.49 +/- 0.01	0.44 +/- 0.03	0.48 +/- 0.02
SW+LW	0.49 +/- 0.09	0.44 +/- 0.04	0.49 +/- 0.02

Table A. Bar length derived from isophotal fitting and Fourier decomposition of the LW and SW+LW images. $R_{\text{bar},1}$ is derived from the FWHM of the I_2/I_0 modes (Ohta et al. 1990), $R_{\text{bar},2}$ is derived from the bar/interbar intensity contrast (Aguerri et al. 2015), and $R_{\text{bar},3}$ is derived from the first minimum after the peak of the ellipticity profile (deprojected image; Wozniak et al. 1995).

It is worth noticing that the Fourier decomposition (supported by the morphological modeling) allows us to rule out the possibility that the elongated component detected by the isophotal analysis could possibly be misled by spiral arms departing from a compact bulge, since these latter could not mimic the $m=2$ peak (see e.g. Figure 1 in Rosas-Guevara et al. 2022). This is now commented on in the main text (lines 54-57).

Finally, the quality of HST images is not sufficient to detect the bar structure in ceers-2112, both for targeting the UV-optical rest-frame wavelength range and for the poor angular resolution of WFC3 images (see Figure A). As for the NIRCcam images (Extended Data Figure 1), we report the resolution of the images as $2 \times \text{FWHM}$ of the PSF, since we assume that bars with projected angular sizes less than twice the PSF FWHM are not detected (see e.g., Erwin et al. 2018; Rosas-Guevara et al. 2022).

Figure A. Postage stamps of ceers-2112 in all HST filters used in this work. We report the angular resolution as $2 \times \text{FWHM}$ of the PSF.

3) Given the above reservations, some of the statements on the paper are currently overstated or not proven. E.g., search for "robust" and "the first", and these need to be reworded more conservatively.

We reword the text accordingly to the new analysis provided. The combination of the three different diagnostics for the bar characterization on the combined image (SW+LW), together with the additional isophotal analysis on individual bands, and the Fourier decomposition on the combined SW and LW images show how ceers-2112 hosts a (prominent $I_2/I_0 > 0.4$) stellar bar (strength $S = 0.23 \pm 0.1$, calculated as in Athanassoula & Misiriotis 2002).

Other comments:

Fig. 1: The central wavelength of the 0.8 μm data point certainly, and perhaps also those of some of the 1.1-1.6 μm data points do not seem to match the filter transmission curves plotted.

Solved. The datapoints were correctly displayed, but the filter transmission curves were not properly normalized to match the limits of the Figure (see new Extended Data Figure 8).

Referee #2 (Remarks to the Author):

This manuscript presents the discovery of a stellar bar structure in a Milky Way progenitor mass ($\sim 10^9 M_{\text{sun}}$) galaxy at $z \sim 3$. This result of a stellar bar structure in a lower mass galaxy within the first 2 Gyr of the Universe is novel and timely, providing important insights into both the properties of high redshift galaxies (that stellar disks can be dynamically cool and form bars) and the formation and evolution of galaxies over time, including our own Milky Way (as stellar bars can help to funnel gas inwards which is necessary for the growth of bulge structures and central super massive black holes).

While the topic of study is novel and timely, some of the analysis does not robustly and unambiguously support the specific stated conclusions. With revision including extended analysis or explicit presentation of existing alternative analysis, this manuscript certainly merits reconsideration for publication.

Major concerns:

1. SFHs: (Sections 3.1.2 & 3.1.3):

The inferred SFH (both integrated and resolved) for ceers-2112 is a critical component of the conclusion that this galaxy rapidly formed a bar in < 400 Myr. However, it is not clear that the conclusion of rapid bar formation is robust. The fiducial integrated fit (using FAST) is performed using an exponentially declining SFH, which may or may not be appropriate for this system and does place prior constraints on the measured SFH (eg, Carnall et al. 2019, doi:10.3847/1538-4357/ab04a2). Why was an exponentially declining SFH adopted, compared to a delayed exponentially declining SFH as with the Synthesizer fitting? Moreover, the full set of fits performed (including both parametric and non-parametric SFHs) exhibit a wide range of star formation timescales. These uncertainties in the recovered SFH imply that the bar formation timescale is also uncertain. Nonetheless, even if the bar formation timescale is more extended or uncertain (from ~ 400 Myr - 1 Gyr), the early epoch and low stellar mass of this galaxy still makes the bar structure remarkable.

We agree with the referee that the stellar population modeling is crucial to infer the SFH of ceers-2112 and discuss its structural evolution. Thus, we provide a more detailed description of our choices, reshaping the main text to lead the reader through our main conclusions. Moreover, thanks to the referee's comment, we revise all the assumptions made for the SFH modeling and realize that our configuration for the non-parametric SFH derived with Prospector was not ideal for describing the SFH of ceers-2112. We revise all the assumptions and adopt a Chabrier IMF, consistent with all the other models, and increase Nbins from 3 to 5, following the prescriptions in Leja et al. 2019 (see also Tacchella et al. 2022). This allows us to provide a more comprehensive description of the early SFH of ceers-2112, consistent with the picture provided by other codes.

Our choice is to model the galaxy using different codes and assumptions to provide a more complete view of possible systematics in deriving the SFH of ceers-2112. In this sense, the exponentially declining SFH is the most basic parametrization. Following the referee's suggestion, we perform a fit using a delayed exponentially declining SFH with Prospector, finding that the galaxy has stellar mass $M = 4.9 \times 10^9 M_{\odot}$ and formation timescale $\tau = 200$ Myr, very consistent with the values obtained with other codes and assumptions.

It is worth noticing that the Synthesizer fit provides a spatially-resolved view of the stellar population properties, allowing the description of a complex SFH, as is the case of ceers-2112, as the sum of multiple delayed exponentially declining SFHs. We see that ceers-2112 has two main star-forming episodes, in quite good agreement with the (integrated) non-parametric fits (Prospector and Dense Basis): the first one could be identified with the buildup of the stellar disk and the second one with the stellar bar growth. We agree with the referee that the timescale of bar growth is quite uncertain, so we provide a median

estimation of ~ 400 Myr by comparing all the SFHs shown in the new Extended Data Figure 9.

To simplify the discussion in the main text, especially for non-expert readers, and considering the spatially-resolved stellar population analysis could be better suited for describing the complex SFH of ceers-2112, we present our findings obtained with synthesizer (2D delayed-tau model), but we add all the details about possible systematics in the stellar population modeling in Methods (Stellar population properties, lines 343-373), Extended Data Figure 9, and Extended Data Table 1.

2. Bar structure measurements: (Section 3.2):

(a) Sections 3.2.2-3.2.4:

The details of the stacked images used for the Fourier analysis and morphological modeling is never stated. Presumably all available JWST/NIRCam bands are used in the stack (F115W, F150W, F200W, F277W, F356W, F410M, F444W), but this is not explicitly stated and should be clarified.

If this is the case, this stack covers from the rest-UV to rest-NIR, and thus dust attenuation and spatially-variable younger stellar populations may result in a composite light distribution that does not follow the full stellar distribution. The authors are certainly aware of this issue, and likely used the stack because of signal-to-noise considerations required for the detailed structural analysis.

However, if at all possible given the imaging depth, it would be interesting to see the detailed structural analysis performed on stack of only the redder bands (F277W and redder), which would mitigate many of the concerns above by probing only the redder portions of the rest optical and NIR while also increasing the composite image depth.

If this is not possible, the potential impact of such factors should be explicitly discussed in the text.

We thank the referee for allowing us to clarify this point. For the morphological analysis, we use a combined image obtained stacking all seven PSF-convolved images. This is now clearly stated in the main text. Furthermore, we add all the details and references about PSF-matching and stacking procedure in Methods (Data, lines 202-208).

We follow the referee's suggestion and create two new combined images, one stacking three SW bands (F115W, F150W, F200W; convolved both to F200W and to F444W) and one stacking three LW bands (F277W, F356W, F444W; convolved to F444W).

We add to this report the isophotal analysis of "stack SW" and "stack LW" images (Figure A; see also Extended Data Figure 6). We see a similar trend at short and long wavelengths, with two main differences: (1) the position angle is almost constant in the "stack LW" image, while it presents more variation in the "stack SW" image (especially the one convolved to F200W), making the identification of the bar less clear in the UV-optical restframe regime; (2) the S/N, especially in the inner and outer regions, is very low in the "stack SW" image with respect to the "stack LW" image. This is now clarified in Methods (Morphology, lines 300-304).

Figure A. (from left to right) Isophotal analysis of ceers-2112 on the stack SW (convolved to F200W), on the stack SW (convolved to F444W), and on the stack LW (convolved to F444W).

Furthermore, we repeat the Fourier analysis on the “stack SW” and “stack LW” images and discuss the results in the new Extended Data Figure 6. We see that the bar component is clearly detected in the “stack LW” (m=2 even component stronger than any other component), while the “stack SW” image shows both prominent m=1 and m=2 components. This is due to the non-asymmetry (lopsidedness) of the elongated structure seen at short wavelengths. Again, while the evidence for the bar structure is present both at short and long wavelengths, the bar is more evident in the “stack LW” image. The bar length derived from isophotal fitting and Fourier analysis in the “stack SW” and “stack SW+LW” are totally compatible (see Table A hereafter). For these reasons, we base our main analysis on the stack “SW+LW” image. We add this discussion in Methods (Morphology, lines 304-311).

	$R_{\text{bar}, 1}$ (arcsec)	$R_{\text{bar}, 2}$ (arcsec)	$R_{\text{bar}, 3}$ (arcsec)
LW	0.49 +/- 0.01	0.44 +/- 0.03	0.48 +/- 0.02
SW+LW	0.49 +/- 0.09	0.44 +/- 0.04	0.49 +/- 0.02

Table A. Bar length derived from isophotal fitting and Fourier decomposition of the LW and SW+LW images. $R_{\text{bar}, 1}$ is derived from the FWHM of the I_2/I_0 modes (Ohta et al. 1990), $R_{\text{bar}, 2}$ is derived from the bar/interbar intensity contrast (Aguerri et al. 2015), and $R_{\text{bar}, 3}$ is derived from the first minimum after the peak of the ellipticity profile (deprojected image; Wozniak et al. 1995).

It is worth noticing that with our two-dimensional stellar population analysis we are able to create the mass density distribution map obtained from the best-fitted spatially-resolved SED, which mimics the barred structure in ceers-2112 (see Figure 2, panel c). This result provides an additional (and independent) confirmation of the reliability of our morphological analysis.

(b) Section 3.2.1: While F200W does probe the rest frame optical at $z \sim 3$ and provides very high spatial resolution, the isophotal analysis could be strengthened by also performing isophotal fits on the F444W band, which probes the stellar continuum and thus provides a much more direct constraint on the stellar distribution in this galaxy.

We thank the referee for this suggestion and repeat the isophotal analysis on six individual bands, three SW (F115W, F150W, F200W) and three LW (F277W, F356W, F444W). The results are discussed in Extended Figure 5. We see that an elongated structure (ellipticity > 0.4) is evident in the LW bands, while the signature for the bar is less clear (but still present) at shorter wavelengths. In particular, the PA measured on individual SW images is not constant at small radii, while it is almost constant if measured on individual LW images (as for the “stack SW” and “stack LW”, see the answer to point 1, Figure A). Furthermore, in SW individual images, the stellar disk is not detected up to F200W.

For these reasons, we based our main analysis on the “stack SW+LW” image. We prove that the analysis of the “stack LW” image confirms our findings based on the “stack SW+LW” image (see also previous point).

Minor comments:

Abstract, line 17, and elsewhere in the text:

This work shows dynamically cold stellar disks for *some* galaxies can be in place *by* $z=4-5$ (depending on the exact inferred SFH), but does not address earlier epochs, nor does this analysis address the universality of cold stellar disks at this epoch and stellar mass.

We agree with the referee that our statement was too generic (and not supported by our work), changing the text accordingly.

Data / Sec 2:

Lines 72-75:

It appears the publicly-available reduced NIRCcam mosaics from the CEERS team were used for this analysis given the references, but this should be explicitly stated.

This is now stated in “Availability of data and material” and a link is provided:

“This study used CEERS JWST/NIRCcam data, which are publicly available from the Mikulski Archive for Space Telescopes (MAST; <http://archive.stsci.edu>) under program ID 1345. Calibrated NIRCcam data products from the CEERS team are available at <https://ceers.github.io/releases.html>.”

Line 76:

Please list the HST filters which were included.

This information is now available in Methods (Data, lines 209-213):

“For studying the stellar population properties of ceers-2112, we extended the NIRCcam wavelength baseline with HST images (F606W, F814W, F125W, F140W, and F160W) from the CANDELS collaboration [39, 40].”

Figures:

- Figs 1 & 2: Image postage stamps and resolved property maps

No size bars have been included, which makes it difficult for the reader to evaluate the physical scale of the galaxy and how the various photometric apertures and structural analysis profiles compare to the observed galaxy extent.

The cutout size is now provided in the new Figure 1. All cutouts in the manuscript have the same size (53x53 px², 1.59x1.59 arcsec², 12.5x12.5 kpc² at z=3.03), which is now reported in the caption of each figure.

- Fig 1: The contours in the multiwavelength postage stamps should be defined

Done.

Results / Sec 3:

Lines 81-83:

How was this galaxy identified as barred for this analysis, given it was not labeled as barred in the visual inspection from Kartaltepe et al. 2023?

The galaxy showed a “peculiar” visual morphology, but the information and quality of the image stretching in individual bands (gray-scale colorbar) available in the .jpg used to classify the sample galaxies did not provide conclusive evidence to visually classify the galaxy as barred (Kartaltepe, private communication). Due to the “peculiar” morphology of ceers-2112, only the follow-up analysis presented in this manuscript, mainly based on the combined stack image (either LW or SW+LW), reveals the presence of the stellar bar component.

Sec 3.1, lines 86-90:

What is the ellipticity of the small photometric apertures, and how was this determined?

How much of the galaxy's structure is captured within these small apertures?

What are the radii and ellipticity of the large apertures?

The normalization procedure is ambiguous as phrased: was the integrated photometry measured on a single band and then the same correction factor applied to all the 0.44" radius color aperture photometry?

Exactly which set of photometric measurements were used in the integrated SED fits: a set of full-band larger aperture photometry, the normalized smaller aperture photometry, or some other case?

Is the nearby extended source a foreground object?

The flux and ellipticity are measured with the rainbow pipeline (see Pérez-Gonzalez et al. 2008; Barro et al. 2011 for all details). The small apertures enclose the inner region of the galaxy (up to the extent of the bar component), while the larger apertures enclose the flux in the outskirts.

The final photometry is based on the flux measured on small apertures (since colors are fundamental to derive the photometric redshift), normalized using the median difference of the flux measured in small and large apertures in all bands.

The nearby source is a foreground extended galaxy ($z_{\text{phot}}=1.1$) with a distance (centroids) of ~ 3.5 arcsec.

All these details are now clarified in the manuscript (Method, Redshift estimation, lines 324-332):

“We carefully measure the ACS, WFC3, and NIRCam photometry with the rainbow code [57, 58], using small elliptical apertures (radius of 0.44 arcsec; $e=0.35$) to retrieve reliable colors and to avoid possible photometric contamination by a foreground extended source ($z_{\text{phot}}=1.1$; projected distance of ~ 3.5 arcsec). Then, we measure the photometry on slightly larger apertures (radius of 0.84 arcsec) to obtain the integrated emission. Finally, we normalize the spectral energy distribution measured on small apertures using the median difference of the flux measured in the small and large apertures.”

Line 101:

“... show that ceers-2112 accreted the bulk of its stars ...” Formed the bulk of its stars? Or is there evidence for interactions through which stars could have been accreted?

We rephrase our sentence (line 65) as:

“mass-weighted age of $620+150-160$ Myr”

Lines 101-102:

These timescale constraints are not necessarily universal, even for this epoch and galaxy stellar mass.

We agree with the referee and modify our statement (lines 68-71) as:

“This analysis suggests that the stellar disk of ceers-2112 assembled at $z\sim 5$ and that the bar component formed 200 Myr later, assembling in ~ 400 Myr, providing a first observational hint on the formation timescale of bars and spiral structures at these early times.”

Lines 102-104:

How do the stellar masses and SFRs from the alternative SED fits compare to the quoted values from FAST?

We provide the masses and SFRs derived from different codes and assumptions in the Extended Data table 1. The uncertainties in the stellar mass estimations are reflected in the error bar shown in the new Figure 3.

	FAST	PROSPECTOR	SYNTHESIZER	PROSPECTOR	DENSE BASIS
SFH	Tau model	Delayed-tau model	Delayed-tau model	Non-parametric	Non-parametric
$M (M_{\odot})$	2.6×10^9	4.9×10^9	3.9×10^9	4.5×10^9	4.6×10^9
$\log(\text{SFR}_{50}) (M_{\odot} \text{ yr}^{-1})$	-0.01	0.33	-0.54	0.73	0.31

Table B. SFHs models, stellar masses, and SFRs derived from the different codes and assumptions used in this work.

Sec 3.2:

As before, the filters included in the stacked image should be stated.

Done.

Sec 3.2.2:

How was the deprojection for the Fourier analysis performed? How was the inclination of the system determined?

The deprojection for the Fourier analysis is now described in Methods (Morphology, lines 246-251):

“To project the galaxy into the face-on view keeping the flux preserved, the image is stretched along the disk minor axis by a factor of $\cos(i_{\text{disk}})^{-1}$, where i_{disk} is the disk inclination derived from the disk ellipticity. In particular, from the isophotal fitting of the combined stack image (SW+LW) we derive $\text{ell}_{\text{disk}} = 0.23$ ($i_{\text{disk}} = 41^\circ$), taking the median values in the outer isophotes ($0.6 < r < 0.7$ arcsec), where the influence of the bar is negligible.”

We further tested our findings by repeating the Fourier decomposition on the “stack SW+LW” image assuming both different position angles (delta PA +/- 5 degrees) and inclinations (delta i +/- 5 degrees) for the galaxy (eight different configurations). The results are summarized in Figure B attached to this report. We see quite consistent results between the different configurations, proving an additional constraint of the reliability of the Fourier characterization of the bar component.

Figure B. Fourier decomposition of ceers-2112 using different values of the inclination and position angle.

We comment on these tests in Methods (Morphology, lines 257-261):

“We further test our findings by repeating the Fourier decomposition on the stack SW+LW image assuming both different position angles ($\Delta PA \pm 5$ degrees) and inclinations ($\Delta i \pm 5$ degrees) for the galaxy (eight different configurations). No systematics are identified in the bar identification due to galaxy deprojection effects”.

Sec 3.2.3 & Fig 3, bottom center panel:

How were the residual contours ascribed to the spiral versus bar structures?

Contours in new Figure 1 (panel c) show different surface-brightness levels in the residual image. For this reason, the “bar residuals” are enclosed within the “spiral residuals”. We overplot to the residual image different isophotal levels and detected a stronger residual in the “bar” region with respect to the “spiral arms” region, pointing to a complex morphology that is not well-described only by the spiral arms. Since the galaxy can be modeled with a disk+Ferrers model, we identify the stronger residuals with the bar structure. It is worth noting that no “bulge” residuals are present.

Sec 3.2.4 & Fig 3, bottom fourth and fifth panels:

What is masked in these panels and the two-component morphological model fitting? How is that mask determined?

The mask for the two-dimensional bar+disk decomposition was built by growing the spiral arms residuals, excluding the bar region. The regions of low S/N were also masked for illustrative purposes.

This is now commented in the text (Methods, Morphology, lines 284-287):

“Since GASP2D does not fit the spiral arm components, we mask them in order to avoid possible contamination in retrieving the ellipticity and position angle of the bar. The mask for the two-dimensional bar+disk decomposition was built by growing the spiral arms residuals, excluding the bar region.” But, since the old version of the Figure was not clear and the mask was hiding the residuals in the outer region of the image, we replace it with the new Extended Data Figure 7 (panel d), where we mark the break radius of the double-exponential disk model, where the surface-brightness of the model rapidly declines.

Sec 3.2.4, lines 166-167:

How does the two-component fit Ferrers bar length compare to the length inferred from the Fourier analysis?

Comparing different diagnostics to infer the length of the stellar bar is not straightforward, since each method presents some caveats (see e.g., Athanassoula & Misiriotis 2002; Aguerri et al. 2003; Erwin 2005). The bar length derived with different diagnostics is now provided in Method (Morphology, lines 313-318):

“Our morphological analysis provides four independent estimations of the bar length : (1) $R_{\text{bar},1} = 0.49 \pm 0.09$ arcsec, from the outer radius of the FWHM of I_2/I_0 [56]; (2) $R_{\text{bar},2} = 0.44 \pm 0.04$ arcsec, from the outer radius of the FWHM of the bar/interbar contrast [48]; (3) $R_{\text{bar},3} = 0.49 \pm 0.02$ arcsec, from the radius at which there is first minimum after the ellipticity peak [45]; and (4) $R_{\text{bar},4} = 0.42 \pm 0.03$ arcsec, from the Ferrers bar modeling [53].

Sec 4, lines 203-205, and Sec 3.1.3:

For the non-expert reader, it would be helpful to explicitly lay out the logic that rapid gas consumption, as inferred from the SFH, allows for the stellar disk to become dynamically cool and form a bar within a short time afterwards.

We rephrase this concept in the main text (lines 95-102):

“Due to their low entropy, galaxy disks with highly ordered rotation are very sensitive to perturbations. However, high-z galaxies are more gas-rich (and turbulent) than local galaxies [3, 25-27] and gas-rich stellar disks stay near-axisymmetric much longer than gas-poor ones, preventing or delaying the formation of the bar component [e.g., 28]. Since ceers-2112 has a mass-weighted age of ~600 Myr, the high gas fraction usually observed in high-z galaxies could have been rapidly consumed during the stellar disk growth before the stellar bar component could start developing [see also 29].”

Reviewer Reports on the First Revision:

Referees' comments:

Referee #1 (Remarks to the Author):

I have read and reviewed the CEERS $z=3$ bar paper again, and want to thank the authors for taking all my suggestions into account. They make a convincing case in Fig. 4-6 now that a bar is present in this object at $z\sim 3$, and significantly more prevalent at the longer rest-frame wavelengths, as expected for a dynamical structure of this 0.4-0.5 Gyr age at $z=3$. Therefore, their earlier somewhat strongly worded statements are now justified. They carefully discuss the important implications of this finding in the context of how early bar formation may have proceeded in MW type progenitors. The paper is now suitable for publication in Nature as is.

Good luck, great work!

Referee #2 (Remarks to the Author):

The updated manuscript is improved thanks to the authors' changes, clarifications, and extended analysis. My concerns about the SFH and the bar structure measurements and conclusions have been well addressed, as have the other comments, and I now recommend this paper for publication.